# Hemorrhagic Colitis Caused by Everolimus in a Patient with Nonfunctional Pancreatic Neuroendocrine Neoplasms: A Case Report

**DOI:** 10.3390/medicina58030410

**Published:** 2022-03-10

**Authors:** Noriyuki Arakawa, Atsushi Irisawa, Kazuyuki Ishida, Takuya Tsunoda, Yoshiko Yamaguchi, Akane Yamabe, Makoto Eizuka, Shunzo Tokioka, Hiroto Wakabayashi

**Affiliations:** 1Department of Gastroenterology, Takeda General Hospital, Aizuwakamatsu 965-8585, Japan; tsunotaku@takeda.or.jp (T.T.); shunzo.tokioka.1103@gmail.com (S.T.); wakaba@takeda.or.jp (H.W.); 2Department of Gastroenterology, Dokkyo Medical University, Mibu 321-0293, Japan; irisawa@dokkyomed.ac.jp (A.I.); yamaaka0110@yahoo.co.jp (A.Y.); 3Department of Diagnostic Pathology, Dokkyo Medical University, Mibu 321-0293, Japan; ishida-k@dokkyomed.ac.jp; 4Department of Diagnostic Pathology, Takeda General Hospital, Aizuwakamatsu 965-8585, Japan; yyamaguchi@takeda.or.jp; 5Department of Internal Medicine, Division of Gastroenterology, Iwate Medical University, Morioka 020-0023, Japan; m10_makoeizuka@yahoo.co.jp

**Keywords:** everolimus, hemorrhagic colitis, nonfunctional pancreatic neuroendocrine neoplasms

## Abstract

A 60-year-old woman was diagnosed with nonfunctional pancreatic neuroendocrine neoplasm with multiple liver metastases and was administered everolimus. Due to persistent epigastric pain and diarrhea, a colonoscopy was performed on the 14th day after the start of everolimus administration, which revealed small bleeding ulcers in the ileocecal region, transverse colon, and rectum. These adverse effects were attributed to the everolimus; it was immediately discontinued, and the patient’s clinical symptoms and imaging findings improved. We concurred that the administration of calcium channel blockers resulted in the inhibition of everolimus metabolism and the disease onset. The everolimus was discontinued. There was no subsequent recurrence of hemorrhagic colitis.

## 1. Introduction

Everolimus, sunitinib, streptozotocin, and lanreotide acetate are all antineoplastic agents for treating nonfunctional pancreatic neuroendocrine neoplasms with synchronous distant metastases. The WHO classification of pancreatic and gastrointestinal neuroendocrine tumors was revised in 2019, resulting in revisions to Japan’s clinical guidelines of pancreatic and gastrointestinal neuroendocrine tumors. These revised guidelines increased the treatment options for the disease [1]. In this report, everolimus was used to treat a case of nonfunctional pancreatic neuroendocrine neoplasm with multiple liver metastases that resulted in drug-induced hemorrhagic colitis. None of the studies in the literature (within the scope of our search) have reported this effect, which should be considered as one of the possible adverse effects of everolimus.

## 2. Case Presentation

A 60-year-old woman was referred to our hospital in early April 2020 with intermittent epigastric pain that began in March 2020, after which she experienced a worsening malaise and loss of appetite. Blood sampling data from the first visit (Table 1) revealed a significant inflammatory response (WBC 11.10 × 10^3^/μL and CRP 3.33 mg/dL); however, no other remarkable findings were observed. An abdominal contrast-enhanced computed tomography (CT) scan revealed a lobulated tumor with early deep staining in the tail of the pancreas. The right hepatic lobe was lobulated and showed multiple heterogeneous tumors with internal necrosis (Figure 1A).

A fine-needle aspiration biopsy was performed under endoscopic ultrasound and revealed a pancreatic neuroendocrine tumor upon histologic examination (equivalent to G2). We decided that surgical resection was contraindicated for this patient, and she was treated with the molecular-targeted drug everolimus [2,3]. Because the patient was diagnosed with hypertension on admission, she was also started on amlodipine besylate (5 mg/day).

The administration of everolimus (10 mg/day) started in the latter half of April 2020. However, on the 14th day of treatment, the patient started complaining of a sharp, intermittent epigastric pain, as well as the passage of tomato juice-like bloody stool, and on the 15th day of administration, bloody diarrhea was observed. Inpatient treatment was started in early May 2020. An abdominal CT scan revealed edematous changes from the ileocecal region to the transverse colon (Figure 1B). A colonoscopy performed on the second day of hospitalization showed redness, erosions, and small ulcers in the ileocecal region, transverse colon, and rectum (Figure 2A–C).

The patient tested negative for *Clostridium difficile* toxins, and no other causes of diarrhea were discovered in the stool culture. Histopathological examination showed diffuse, moderate inflammatory cell infiltration (mainly lymphocytes) and apoptotic bodies in the crypt epithelium (Figure 3A,B).

There were no crypt abscesses, granulomas, amyloid deposits, or nuclear inclusions, and no other specific inflammatory or malignant findings. On the fourth day, following the withdrawal of the calcium channel blocker and the everolimus, the abdominal symptoms and melena disappeared. The patient was discharged from the hospital on the seventh day after admission. The blood trough level of everolimus was 94.4 ng/mL (standard value; 5–15 ng/mL). The patient was diagnosed with everolimus-induced hemorrhagic colitis—based on the clinical course and laboratory findings—which was possibly caused by an increased blood concentration of everolimus due to amlodipine besylate. A CT scan of the abdomen and a colonoscopy performed (as part of the follow-up) in early June showed the disappearance of the inflammation in the large intestine. Somatostatin receptor scintigraphy showed uptake in the primary pancreatic and liver metastases; therefore, we investigated the addition of lanreotide acetate. The administration of lanreotide acetate (120 mg, once every four weeks) was started. The course was continued without any relapse of hemorrhagic colitis. In terms of the antitumor effect, the disease had been stable for 24 months after the initial diagnosis when assessed at a follow-up appointment.

## 3. Discussion

The mammalian target of rapamycin (mTOR) is a serine/threonine kinase that regulates various functions such as cell proliferation, metabolic activity, angiogenesis, and autophagy control and plays a central role in the PI3K/AKT signal transduction pathway [4]. Everolimus is an mTOR inhibitor that forms a complex that binds to mTOR. As a result, the drug inhibits cell proliferation and suppresses tumor cell proliferation. Furthermore, the inhibition of angiogenesis suppresses tumor growth. This signal transduction pathway has been found to be activated in patients with malignant tumors [5]. In Japan, this drug was approved for the radical treatment of unresectable renal cell carcinoma and metastatic renal cell carcinoma in 2010, pancreatic neuroendocrine neoplasms in 2011, and inoperable and recurrent breast cancer in 2014. The advent of molecularly targeted drugs has dramatically improved the prognosis [6].

An international, jointly conducted, double-blind, placebo-controlled phase III trial (RADIANT-4 trial) compared everolimus to placebos in patients with nonfunctional advanced lung or gastrointestinal neuroendocrine tumors (NETs), and associated adverse events were reported [7]. The most common adverse events were stomatitis (63%), diarrhea (31%), fatigue (31%), infections (29%), rashes (27%), peripheral edema (26%), and non-infectious pneumonia (including interstitial pneumonia) (16%). We could not find any reports that described hemorrhagic colitis as an adverse effect of everolimus, as shown in the present case. By searching the literature for molecularly targeted drugs, we confirmed one report on sorafenib tosylate with a similar adverse effect. There have been several reports of everolimus-induced gastrointestinal bleeding. These reports suggest some relationship between mTOR inhibitors and gastrointestinal bleeding. The mTOR inhibitors may prevent mucosal healing in the stomach and trigger gastrointestinal bleeding [8].

In this study, we discussed the mechanism of gastrointestinal disorders associated with everolimus. It has been reported that mTOR activates the PI3K/AKT signal transduction pathway and promotes angiogenesis through vascular endothelial growth factor (VEGF) [9]. A VEGF inhibitor induces gastrointestinal disorders by destroying both tumor and non-tumor blood vessels, which results in gastrointestinal bleeding [10]. The mechanism of thromboembolic formation and microcirculatory disorders have also been reported [11]. It is thought that inhibiting mTOR could cause gastrointestinal disorders through the same mechanism as VEGF inhibitors.

We differentiated between infectious enteritis, inflammatory bowel disease (ulcerative colitis and Crohn’s disease), drug-induced hemorrhagic enteritis, ischemic enteritis, vasculitis-related diseases, and cytomegalovirus enteritis by using CT scans and colonoscopy findings; we also conducted detailed examinations. The colonoscopy biopsy resulted in no specific findings on the histopathological examination. Blood sampling showed no abnormal markers for vasculitis-related diseases; the stool culture results showed no obvious pathogens. The patient had no history of oral antibiotic use prior to the onset of symptoms; therefore, our final diagnosis was everolimus-induced hemorrhagic colitis. The oral administration of a calcium channel blocker, which is an antihypertensive drug, was thought to have enhanced the onset of the symptoms. Everolimus is primarily metabolized by the hepatic metabolizing enzyme CYP3A4. It is speculated that the suppression of CYP3A4 by the calcium channel blocker, in turn, inhibited everolimus metabolism, resulting in an increased blood concentration of everolimus. It is assumed that the concomitant use of both drugs is the cause of this side effect.

## 4. Conclusions

In conclusion, we encountered a case of hemorrhagic colitis caused by everolimus in our practice that had not been previously reported in the literature. It is better to avoid prescribing drugs with a CYP3A4 inhibitory effect combined with everolimus to avoid adverse reactions.

## Figures and Tables

**Figure 1 medicina-58-00410-f001:**
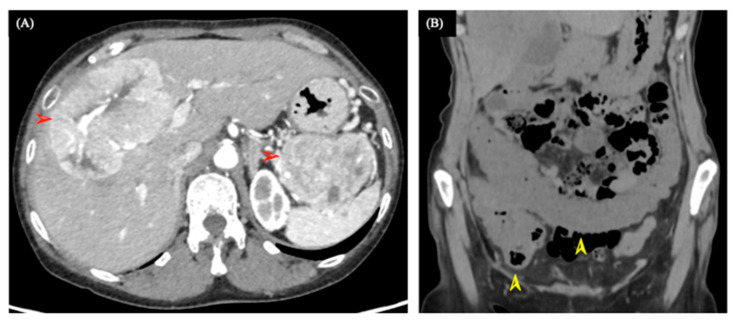
(**A**) Abdominal contrast-enhanced computed tomography (CT) during the first visit: a tumor with early dye uptake is seen in the tail of the pancreas and right lobe of the liver (red arrow) with areas of necrosis. (**B**) An abdominal CT scan during the occurrence of melena: the coronal section showing edematous changes observed from the cecum to the transverse colon (yellow arrows).

**Figure 2 medicina-58-00410-f002:**
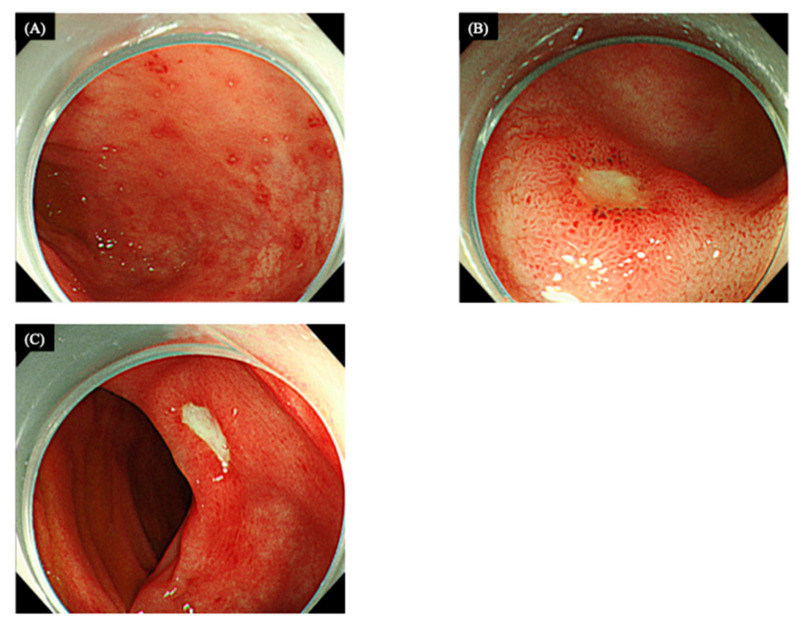
(**A**) Multiple superficial erosions surrounded by hyperemic mucosa in the rectum. (**B**) An ulcer surrounded by hyperemic edematous mucosa in the terminal ileum. (**C**) Ulcers with well-defined rounded white moss seen in the mucosa of the transverse colon and accompanied by edematous changes in the background.

**Figure 3 medicina-58-00410-f003:**
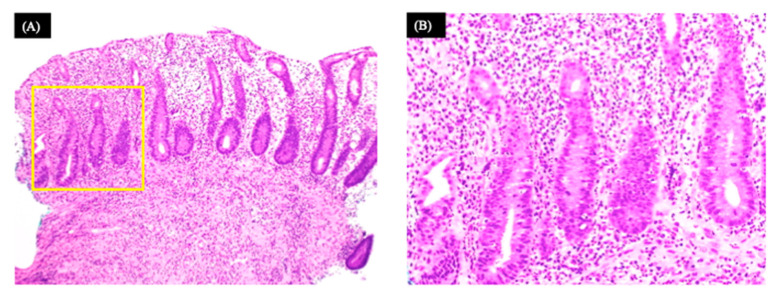
(**A**) A histopathological biopsy image under low magnification showing diffuse moderate inflammatory cell infiltration (mainly lymphocytes). (**B**) High magnification of the area in the yellow frame. Apoptotic bodies observed in crypt epithelium.

**Table 1 medicina-58-00410-t001:** Initial laboratory findings of the patient.

Hematology				Serology	
WBC	11.10 × 10^3^/μL	ALP	209 U/L	CRP	3.33 mg/dL
RBC	485 × 10^4^/μL	γGTP	64 U/L	IgG	1182 mg/dL
Hb	14.5 g/dL	LDH	209 U/L	IgA	160 mg/dL
Ht	41.9%	Na	138 mEq/L	IgM	436 mg/dL
PLT	29.6 × 10^4^/μL	K	4.5 mEq/L	**Other**	
PT	99%	Cl	101 mEq/L	ANA	<40
APTT	33.9 sec	Ca	9.3 mEq/L	ANCA(C/P)	<1.0 EU
**Biochemistry**		TP	7.5 g/dL	CEA	2.7 ng/mL
T-Bill	0.4 mg/dL	Alb	3.8 g/dL	CA19-9	9 U/mL
AST	19 U/L	Bun	10.8 mg/dL	Insulin	10.0 μIU/mL
ALT	19 U/L	Cre	0.74 mg/dL	Glucagon	11.0 pg/mL

## Data Availability

The data that support the findings of this study are available from the corresponding author, N.A., upon reasonable request.

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
