# Peer review of "Hemorrhagic Colitis Caused by Everolimus in a Patient with Nonfunctional Pancreatic Neuroendocrine Neoplasms: A Case Report"

_medicina, 2022, doi:10.3390/medicina58030410_

Round 1
Reviewer 1 Report
I suggest authors should include in the discussion information on other cases of gastro-intestinal hemorhage caused by everolimus: such as (but not limited to if more present):
- Tsunematsu et al., Severe gastrointestinal hemorrhage related to everolimus: a case report Clin J Gastroenterol. 2019 Dec;12(6):552-555. doi: 10.1007/s12328-019-00978-8.
- Assi & Abdel-Samad Severe gastrointestinal hemorrhage during targeted therapy for advanced breast carcinoma. Curr Oncol. 2014;21:e732–5.
- Gonzales et al. Everolimus implicated in case of severe gastrointestinal hemorrhage. Case Rep Oncol Med. 2017;2017:3657812.
This may increase the overall interest to the readers by providing reference to other cases of GI hemorhage related with everolimus use.
Reviewer 2 Report
This is a case report of drug-induced toxicity between both everolimus and calcium channel blocker; amlodipine.
I have the following comments:
Case presentation:
Figure 1 A: ....with early deep staining better to be changed to early dye uptake.
Necrosis in the interior... better to be changed to with areas of necrosis.
Figure 2 A: Reddish erosions in multiple areas of the rectum .... better to be multiple superficial erosions surrounded by hyperemic mucosa in the rectum.
Figure 2 B: Small ulcers with white moss in the ileocecal region with mild peripherally located edemas ...better to change to an ulcer surrounded by hyperemic edematous mucosa in the terminal ileum.
On the fourth day, following the withdrawal of the calcium channel blocker, the abdominal symptoms and melena disappeared......It is not clear whether you stop both medications or amlodipine only and why?
The blood trough level of everolimus was 94.4 ng/mL......What the trough value of the everolimus should be? To be written between brackets.
diagnosed with everolimus......diagnosed as.
The patient was discharged from the hospital on the seventh day after admission.......What are the discharge medications the patient used?
Everolimus was discontinued......Why you stop everolimus? If you stopped amlodipine alone, this could be enough as it is not recorded that everolimus alone causes this complication.
Discussion:
The advent of molecularly targeted drugs has dramatically improved vital prognosis ......improved the prognosis.
Considering that there was no recurrence after discontinuing everolimus, it is assumed that this was a dose-dependent side effect......It is not an accurate expression but it is merely a drug-drug interaction that causes the hemorrhagic colitis i.e if you use everolimus alone in the usuall therapeutic dose, you may not get this effect so better to write" the concomittent use of both drugs is the cause of this side effect".
Conclusion:
When prescribing drugs with a CYP3A4 inhibitory effect, changing or discontinuing the drug should be considered if there are adverse reactions. .. better to be more accurate " It is better to avoid prescribing drugs with a CYP3A4 inhibitory effect combined with everolimus to avoid adverse reactions.
